# LangSplatV2: High-dimensional 3D Language Gaussian Splatting with 450+ FPS

**Wanhua Li**[1,*]  **Yujie Zhao**[1,2,*]  **Minghan Qin**[3,*]  **Yang Liu**[3]  **Yuanhao Cai**[4]
**Chuang Gan**[5,6]  **Hanspeter Pfister**[1]

[1]Harvard University  [2]University of Chinese Academy of Sciences  [3]Tsinghua University
[4]Johns Hopkins University  [5]MIT-IBM Watson AI Lab  [6]UMass Amherst

## Abstract

In this paper, we introduce LangSplatV2, which achieves high-dimensional feature splatting at 476.2 FPS and 3D open-vocabulary text querying at 384.6 FPS for high-resolution images, providing a 42 × speedup and a 47 × boost over LangSplat respectively, along with improved query accuracy. LangSplat employs Gaussian Splatting to embed 2D CLIP language features into 3D, significantly enhancing speed and learning a precise 3D language field with SAM semantics. Such advancements in 3D language fields are crucial for applications that require language interaction within complex scenes. However, LangSplat does not yet achieve real-time inference performance (8.2 FPS), even with advanced A100 GPUs, severely limiting its broader application. In this paper, we first conduct a detailed time analysis of LangSplat, identifying the heavyweight decoder as the primary speed bottleneck. Our solution, LangSplatV2 assumes that each Gaussian acts as a sparse code within a global dictionary, leading to the learning of a 3D sparse coefficient field that entirely eliminates the need for a heavyweight decoder. By leveraging this sparsity, we further propose an efficient sparse coefficient splatting method with CUDA optimization, rendering high-dimensional feature maps at high quality while incurring only the time cost of splatting an ultra-low-dimensional feature. Our experimental results demonstrate that LangSplatV2 not only achieves better or competitive query accuracy but is also significantly faster. Codes and demos are available at our project page: `https://langsplat-v2.github.io`.

## 1 Introduction

Seamless and intuitive interactions between humans and complex 3D environments [1, 2] are paramount for a wide range of applications, such as augmented reality [3, 4] and intelligent robotics [5, 6, 7]. To achieve such interactions, systems must understand and respond to natural language queries in real time. This capability hinges on advancements in 3D language fields—a technology at the intersection of vision-language models [8, 9] and 3D environmental modeling [10, 11].

LangSplat [12], a pioneering model in this domain, leverages 3D Gaussian Splatting to embed 2D CLIP language features into 3D spaces. This method has significantly accelerated the 3D query time, achieving speeds up to 199 × faster than its predecessors [13], which is critical for applications in scenarios where rapid response is essential. Although many recent improvements have been proposed [14, 15], the inference speed remains a major concern, especially at high resolutions, which hinders their widespread application.

Real-time querying is crucial for applications such as real-time navigation [6], interactive gaming [16], and on-the-fly educational tools [17], where delays can disrupt user experience and functionality. To

---

*Equal contribution.

39th Conference on Neural Information Processing Systems (NeurIPS 2025).

identify the speed bottleneck of LangSplat, we decompose the entire querying process into three stages: rendering, decoding, and post-processing. Then, we provide a detailed time analysis on the LERF dataset with one A100 GPU. Results in Table 1 show that each query takes 122.1 ms for LangSplat. With some simple engineering modifications, the rendering and post-processing time can be significantly reduced, we term this new version of LangSplat as LangSplat*. The performance of LangSplat* revealed that the decoding stage, which is responsible for transforming low-dimensional latent features into high-dimensional CLIP features with a heavy-weight multi-layer perceptron (MLP), is the primary bottleneck. LangSplat introduces the MLP decoder to significantly reduce the training memory and time cost. Consequently, an additional decoding stage with an MLP is inevitable for test-time querying and reducing the dimensionality of high-dimensional features lowers the accuracy of the queries. However, simply removing the decoder and directly training the high-dimensional CLIP feature for each Gaussian dramatically increases the rendering time. As shown in Figure 1, as the feature splatting dimension increases, the rendering time of LangSplat significantly increases. Specifically, rendering a feature with a dimension of 1536 (assuming we render three semantic levels of 512-dimensional CLIP feature fields in parallel) is 15 times slower than rendering a 3-dimensional field on one A100 GPU.

To address the decoding bottleneck in LangSplat, we introduce LangSplatV2, which drastically enhances querying speed without sacrificing querying accuracy. As shown in Figure 1, our LangSplatV2 successfully decouples rendering speed from the dimensionality of rendering features, enabling the rendering of high-dimensional features at the computational cost of splatting an ultra-low-dimensional feature. As the millions of 3D Gaussian points actually represent a limited number of unique semantics for a 3D scene, our LangSplatV2 assumes that each Gaussian can be represented as a sparse coding over a group of global basis vectors. Then, we derive that learning a high-dimensional feature field is equivalent to learning a 3D coefficient field and a global codebook. For each 3D Gaussian point, instead of augmenting a language feature, we learn sparse coefficients. During testing, we utilize the sparsity and propose an efficient sparse coefficient splatting method with CUDA optimization, which effectively decouples the rendering dimension with

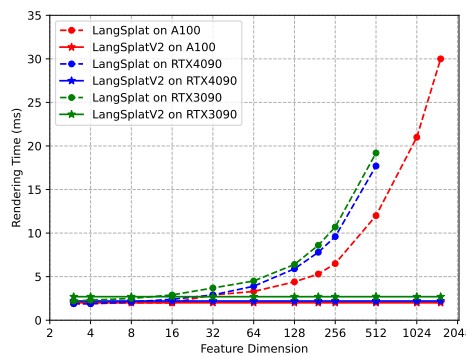

Figure 1: Feature rendering time comparison with different GPUs. Note that the less advanced GPUs (RTX 3090 and RTX 4090) cannot accommodate the LangSplat model with feature dimensions of 1024 or higher due to running out of memory.

feature dimensions. As our method removes the autoencoder and directly learns the 3D field from high-dimensional 2D CLIP features, more accurate language fields are also obtained. Experimental results demonstrate that our LangSplatV2 not only achieves higher accuracy but also delivers a substantial speedup: 42 × faster than LangSplat for high-dimensional feature rendering and 47 × faster for open-vocabulary 3D querying.

## 2   Related Work

**3D Gaussian Splatting.** Recently, 3D Gaussian Splatting (3D-GS) [11] has received huge attention [18, 19] due to its speed advantage [20, 21] over neural radiance fields (NeRFs) [10]. As a fundamental 3D modeling technology, it's been widely used in many downstream tasks [22, 23, 24, 25, 26, 27, 28, 29]. Yang *et al.* [30] extended 3D Gaussian Splatting to dynamic scenes by modeling deformable 3D Gaussians. Concurrently, 4D Gaussian Splatting [31] proposes to use multi-resolution voxel planes to model deformed Gaussians and attains high-quality video rendering results in real-time. Due to the explicit 3D modeling nature of 3D Gaussian Splatting, it has also been used for 3D editing. Gaussianeditor [32] offers swift and controllable 3D editing by tracing the editing target throughout the training process. Another concurrent work [33] edits 3D scenes using text instruction, which attains two times faster training speed compared with Instruct-NeRF2NeRF [34]. Many methods [35, 36, 37] utilize 3D Gaussian Splatting for text-driven 3D generation. GSGEN [38] incorporates direct geometric priors from 3D Gaussians and obtains a text-to-3D generation model that

captures high-frequency components. 3D-GS has also been adopted for generative 3D reconstruction, which has recently attracted significant interest owing to its effectiveness with sparse observations. The first framework of its kind was introduced by Liu *et al.* [39], who pioneered generative 3D reconstruction by modeling it as a temporal generation task. Wang *et al.* [40] subsequently accelerated this approach by using one-step diffusion. Recently, 3D language fields have also embraced 3D-GS. LangSplat [12] adopts 3D-GS to model a 3D language field and is 199 $\times$ faster than LERF [13]. However, LangSplat is still not a real-time method, which undermines its broad application prospects.

**3D Language Field.** The concept of 3D language fields [41, 42, 43, 44, 45, 46] has emerged as an intersection of vision-language models and 3D modeling, aiming to create interactive environments that can be manipulated and queried using natural language. LERF [13] explores 3D open-vocabulary text queries by embedding the CLIP feature into a 3D field. Liu *et al.* [47] also distilling knowledge from pre-trained foundation models like CLIP [8] and DINO [48] into NeRF [10]. It further proposes relevancy-distribution alignment and feature-distribution alignment losses to address the CLIP feature ambiguity issue. LangSplat improves LERF with 3D Gaussian Splatting and attains a significant speedup. There are also some other works [49, 50, 51] employing 3D Gaussian Splatting for 3D language field modeling, they usually suffer from imprecise language fields like LERF, which has been well addressed by LangSplat. Following LangSplat, GOI [52] proposes simplifying feature selection by dividing the feature space with a hyperplane, keeping only the features most relevant to the query. GAGS [15] enhances multiview consistency by linking SAM's prompt point density with camera distances and introducing an unsupervised granularity factor to selectively distill consistent 2D features. However, these methods both use a decoder to map the low-dimensional feature to high-dimensional space, which is a primary bottleneck of real-time query as shown in section 3.2.

**3D Gaussian Splatting Acceleration.** To reduce computational overhead and improve rendering speed, recent works have proposed various strategies to accelerate 3D-GS. One line of work compresses the Gaussian representation using vector quantization and codebooks. For instance, Compressed 3D Gaussian Splatting [53] adopts sensitivity-aware clustering and quantization-aware training to compress both directional colors and Gaussian parameters, achieving up to 31$\times$ compression and 4$\times$ rendering speedup. CompGS [54] similarly applies K-means-based quantization to achieve over 40$\times$ storage reduction with minimal quality loss. Other approaches focus on structural regularization or architectural simplification. Self-Organizing Gaussian Grids [55] reorganize Gaussian parameters into spatially coherent 2D grids, reducing redundancy via local homogeneity. LightGaussian [56] uses pruning based on global significance and vector quantization, achieving 15$\times$ compression and 200+ FPS rendering. C3DGS [57] proposes to reduce the number of Gaussians via a learnable mask and introduces compact view-dependent color encoding, achieving a 25$\times$ reduction in storage. Techniques like Speedy-Splat [58] and SORT-Free Splatting [59] improve runtime performance through optimized rasterization and differentiable approximations of alpha blending. In contrast to these works that focus on accelerating RGB-based scene rendering, our method tackles the unique challenge of handling high-dimensional Gaussian features. Extending prior compression techniques [53, 56] to our setting would require learning and quantizing such high-dimensional attributes per Gaussian, leading to excessive memory usage. Moreover, directly rendering with full 512D features remains computationally demanding. In contrast, our method circumvents the need to directly learn and render high-dimensional features for 3D Gaussian points.

## 3 Proposed Approach

### 3.1 Preliminaries

3D Gaussian Splatting employs point-based rendering technologies [60, 61] and models the scene geometry as a set of 3D Gaussian points. Each Gaussian point is associated with the following attributes: Gaussian center position $\mu \in \mathbb{R}^3$, a covariance matrix $\Sigma \in \mathbb{R}^{3 \times 3}$, color described $c \in \mathbb{R}^3$ by spherical harmonic (SH) coefficients, and opacity $o \in \mathbb{R}$. Each Gaussian point is represented as:

$$G(x) = \exp(-\frac{1}{2}(x - \mu)^\top \Sigma^{-1}(x - \mu)). \tag{1}$$

The covariance matrix $\Sigma$ is further represented with a rotation matrix and a scaling factor matrix:

$$\Sigma = RSS^\top R^\top, \tag{2}$$

where $R \in \mathbb{R}^4$ denotes the rotation matrix and $S \in \mathbb{R}^3$ represents the scaling matrix.

Table 1: The stage-wise time cost (ms) for LangSplat and our improvements on the LERF dataset with one A100 GPU. LangSplat* is modified from LangSplat with simple engineering optimization. LangSplatV2 achieves a speed of up to 384.6 FPS for open-vocabulary 3D querying.

| Method | Rendering | Decoding | Post-Processing | Total | Speed (FPS) |
|---|---|---|---|---|---|
| LangSplat | 6.0 | 83.1 | 33.0 | 122.1 | 8.2 |
| LangSplat* | 2.0 | 83.1 | 0.5 | 85.6 | 11.7 |
| LangSplatV2 | **2.0** | **0.1** | **0.5** | **2.6** | **384.6** |

Based on EWA volume splatting [62], 3D Gaussian Splatting projects the 3D Gaussian points onto a 2D image plane, which blends the ordered Gaussian points that overlap with the rendered pixel $v$:

$$C(v) = \sum_{i \in \mathcal{N}} c_i \alpha_i \prod_{j=1}^{i-1} (1 - \alpha_j), \tag{3}$$

where $\mathcal{N}$ represents the ordered Gaussian points within a tile, and $\alpha_i = o_i G_i^{2D}(v)$. The $G_i^{2D}(\cdot)$ means the projected 2D Gaussian distribution of the $i$-th 3D Gaussian point.

As 3D Gaussian Splatting exhibits excellent real-time rendering speed while maintaining high rendering quality even at 1080p resolution, many methods [31, 30, 63] have adopted it as a 3D scene modeling technology to accelerate the rendering speed. LangSplat aims to build a 3D language field to support open-vocabulary querying within 3D spaces. It extends each 3D Gaussian with a language embedding and supervises the 3D language Gaussians with 2D CLIP image features. LangSplat presents two main contributions to make it faster and more accurate. First, instead of using multiple patch-wise CLIP features with varying scales, which leads to imprecise and vague 3D language fields, LangSplat employs the CLIP features of SAM masks with three pre-defined SAM hierarchical semantic scales to obtain precise language fields with clear boundaries. Second, as CLIP embeddings are high-dimensional, directly splatting at CLIP feature space poses huge challenges in training memory and time cost. Therefore, LangSplat proposes a scene-specific autoencoder, which compresses the high-dimensional (512-D) CLIP features into low-dimensional (3-D) latent features. The rendering process of LangSplat follows:

$$\boldsymbol{F}(v) = \sum_{i \in \mathcal{N}} \boldsymbol{f}_i \alpha_i \prod_{j=1}^{i-1} (1 - \alpha_j), \tag{4}$$

where $\boldsymbol{f}_i \in \mathbb{R}^d$ represents the augmented $d$-dimensional latent feature at $i$-th 3D Gaussian point and $\boldsymbol{F}(v)$ denotes the rendered latent feature at pixel $v$. After obtaining the rendered feature, LangSplat uses the decoder $g_d(\boldsymbol{F}(v)) \in \mathbb{R}^D$ from the trained autoencoder $g$ to decode the latent feature back to CLIP feature space. In the end, the LangSplat attains an accurate 3D language field while being 199 $\times$ faster than the previous state-of-the-art method LERF [13].

## 3.2 Bottleneck Analysis

While LangSplat achieved significant speedup over other methods, it still cannot perform real-time open-vocabulary 3D querying at high resolution. However, there is a high demand for real-time 3D querying for many applications, such as Augmented Reality (AR) [64], intelligent robotics [13].

To further improve the query speed of LangSplat, we first analyze each step of the query process and identify the bottleneck. For a given text query, the inference of LangSplat can be divided into three stages: rendering, decoding, and post-processing. The rendering stage will render the learned 3D language Gaussians into a 2D image plane and get a rendered latent feature map $\boldsymbol{F} \in \mathbb{R}^{H \times W \times d}$, where $H, W$ denote the height and width of the 2D image size, respectively. As the rendered feature map only encodes language features in the low-dimensional latent space, a decoding stage is followed to obtain the feature map at CLIP space with a decoder $g_d(\boldsymbol{F}) \in \mathbb{R}^{H \times W \times D}$, where the decoder $g_d(\cdot)$ is implemented with an MLP. The last post-processing stage computes the relevancy score with the obtained $D$-dimensional feature following LERF [13] and remove some noise in the relevancy score map. Specifically, a mean filter is applied to the relevancy score map. Note that LangSplat adopts the three semantic levels introduced by SAM [65], so the above operations are repeated three times for three SAM semantic levels and the post-processing stage needs to select one semantic level for prediction with some specific strategies [12].

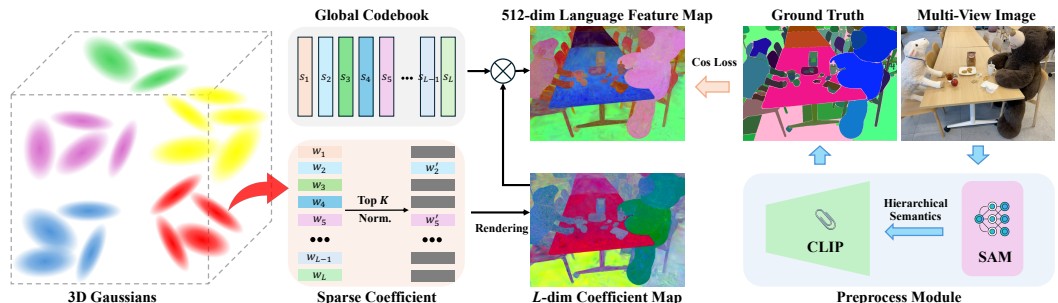

Figure 2: The framework of LangSplatV2. LangSplatV2 introduces a sparse coefficient for each Gaussian point and a shared global codebook for the entire scene.

We test the stage-wise inference time of LangSplat on the LERF dataset with one A100 GPU. The results in Table 1 show that the rendering stage takes 6.0 ms, the decoding stage takes 83.1 ms, and the post-processing stage takes 33.0 ms. While all stages occupy a considerable amount of time, we can significantly improve them through two simple modifications. First, we notice that LangSplat implements the post-processing stage on the CPU, which could be slow when it involves mean filter operations. We implement the post-processing stage on the GPU and observe that the time cost of the post-processing stage is now negligible compared with other stages. The second modification is scale parallelization. LangSplat performs text querying on three semantic levels sequentially, meaning the rendering stage is performed three times separately. However, we can perform inference at three semantic scales in parallel. For example, instead of splatting $d$-dimensional features three times, we directly splat $3d$-dimensional features for once. We use LangSplat* to denote the version with these simple modifications. We observe that for LangSplat*, the rendering stage takes 2.0 ms, and the post-processing stage takes 0.5 ms. In these three stages, the decoding phase occupies 97.1% of the total query time, making it the primary bottleneck in achieving real-time 3D querying.

### 3.3  3D Sparse Coefficient Field

Different from 3D Gaussian Splatting, which renders 3-dimensional RGB color, 3D language fields need to splat high-dimensional features. Directly splatting high-dimensional features significantly decreases the rendering speed, as shown in Figure 1. To address this issue, existing methods usually model $d$-dimensional ($d << D$) latent features in 3D followed by either an online decoder [51] or an offline decoder [12] to decode latent features back to $D$-dimensional features in 2D image space. The high dimension gap between latent features and decoded features implies a heavyweight MLP is required to ensure the accuracy of the decoding stage, which becomes the primary speed bottleneck.

In LangSplatV2, instead of splatting the $d$-dimensional latent language features, we propose to model a 3D sparse coefficient field. As shown in Eq. 4, LangSplat assigns each Gaussian point with a language feature $\boldsymbol{f}_i$, which represents the semantics associated with the Gaussian point. As LangSplat could create millions of Gaussian points, there will be millions of unique language features. However, this is highly inefficient as the unique semantics within a scene are quite limited and much smaller than the number of Gaussian points. As a matter of fact, many Gaussian points share similar semantics. Therefore, we assume the language embedding of every Gaussian point within a scene can be represented as a sparse coding of $L$ global basis vectors $\mathcal{S} = [\boldsymbol{s}_1, \boldsymbol{s}_2, ..., \boldsymbol{s}_L]^\top \in \mathbb{R}^{L \times D}$. These $L$ basis vectors serve as the global codebook and each Gaussian point is computed by linearly combining a small number of local basis vectors. We assume that only $K$ ($K << L$) basis vectors from $L$ global codebook are used to represent the language embedding of a Gaussian point. Then we define $\boldsymbol{w}_i = [w_{i,1}, w_{i,2}, ..., w_{i,L}]^\top \in \mathbb{R}^{1 \times L}$ as the associated sparse coefficients for $i$-th Gaussian point, where $\sum_{l=1}^{L} w_{i,l} = 1$ and only $K$ elements are non-zero values while all other elements are zeros. With slightly abuse of the notion $\boldsymbol{f}_i$, the language embedding $\boldsymbol{f}_i \in \mathbb{R}^D$ of $i$-th Gaussian point is represented as:

$$\boldsymbol{f}_i = \boldsymbol{w}_i \mathcal{S} = \sum_{l=1}^{L} w_{i,l} \boldsymbol{s}_l. \tag{5}$$

If we directly render the $D$-dimensional language field without compressing CLIP features, we have:

$$\boldsymbol{S} = \sum_{i \in \mathcal{N}} \boldsymbol{w}_i \mathcal{S} \alpha_i \prod_{j=1}^{i-1}(1-\alpha_j), \tag{6}$$

where $\boldsymbol{S}$ is the rendered $D$-dimensional CLIP feature. We set $e_i = \alpha_i \prod_{j=1}^{i-1}(1-\alpha_j)$, then we have:

$$\boldsymbol{S} = \sum_{i \in \mathcal{N}} \boldsymbol{w}_i \mathcal{S} e_i = (\sum_{i \in \mathcal{N}} e_i \boldsymbol{w}_i) \mathcal{S}. \tag{7}$$

Eq. 7 shows that rendering the $D$-dimensional CLIP features is equivalent to first rendering the sparse coefficients $\boldsymbol{w}(i)$, and then performing a matrix multiplication with the global dictionary $\mathcal{S}$.

Therefore, we propose to learn a $L$-dimensional sparse coefficient for each Gaussian point and a global codebook with a size of $L$ basis vectors. To learn a sparse distribution for the coefficient $\boldsymbol{w}_i$, we first apply a softmax function to normalize the $L$-dimensional parameter, then preserve the top-$K$ values as non-zeros elements and set the remaining elements to zeros. We will re-normalize the top-$K$ values with their sum to ensure the sum of the coefficient equals one. The $L$-dimensional 3D sparse coefficient field and the $L$ basis vectors are jointly learned. Figure 2 visualizes the framework.

Compared to LangSplat, LangSplatV2 requires only a simple matrix multiplication after rendering the weight map, instead of relying on a heavyweight decoder, thus overcoming LangSplat's main inference speed bottleneck. Furthermore, the $D$-dimensional global codebook eliminates reconstruction loss from dimensionality reduction, enabling better modeling of high-dimensional features in the CLIP space and improving query accuracy.

### 3.4 Efficient Sparse Coefficient Splatting

By learning a 3D sparse coefficient field, the MLP decoder is entirely removed, eliminating the associated computational overhead of the MLP. However, we still need to perform a $3L$-dimensional feature rendering and a matrix multiplication. Our experiments show that the time for matrix multiplication (in Table 1, listed as the decoding stage of LangSplatV2) is negligible compared to the rendering process. However, rendering high-dimensional features with dimensionality $L$ remains computationally demanding. To address this issue, we propose an efficient sparse coefficient splatting method with CUDA optimization. By exploiting the sparsity of the coefficient field, we can achieve $L$-dimensional feature rendering at the cost of only $K$-dimensional rendering, where $K \ll L$.

In the CUDA implementation of 3D Gaussian Splatting and LangSplat, each thread performs alpha-blending of the $|\mathcal{N}|$ ordered Gaussian points within a tile. With an $L$-dimensional rendering, each thread sequentially computes $L$ channels, leading to a computational complexity of $O(|\mathcal{N}|L)$. When $L$ becomes sufficiently large, this alpha-blending computation becomes the key bottleneck and significantly increases the overall computational overhead as $L$ grows, as shown in Figure 1.

To mitigate this, we utilize the sparse nature of the learned coefficient field during test-time querying, as shown in Figure 3. Although each Gaussian point has an $L$-dimensional coefficient, only $K$ dimensions contain non-zero values. Therefore we can only perform alpha-blending

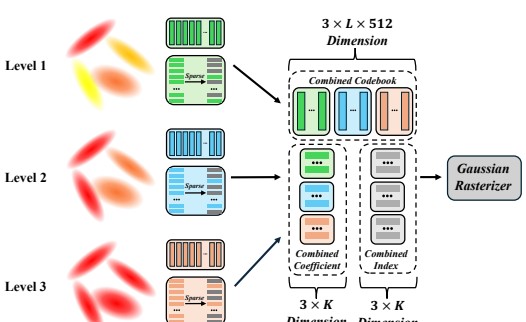

Figure 3: Our efficient sparse coefficient splatting method accelerates the speed of alpha-blending by utilizing the property of the learned sparse coefficient field and neglecting zero elements.

for the non-zero elements, as the blending of zero elements does not alter the result. This effectively reduces the computational complexity from $O(|\mathcal{N}|L)$ to $O(|\mathcal{N}|K)$, with $K$ significantly smaller than $L$. In practice, we set $K = 4$, which allows us to simultaneously render three semantic scales with an effective feature rendering dimension in CUDA of only 12, yielding a high-quality feature map of $1,536$ dimensions. It makes the rendering highly efficient without compromising feature quality.

Table 2: Quantitative comparisons of open-vocabulary 3D object localization and 3D semantic segmentation on the LERF dataset. We report the mean accuracy (%) and the mean IoU scores (%).

| Method | 3D Object Localization | | | | | 3D Semantic Segmentation | | | | |
| --- | --- | --- | --- | --- | --- | --- | --- | --- | --- | --- |
| | Ramen | Teatime | Kitchen | Figurines | Overall | Ramen | Teatime | Kitchen | Figurines | Overall |
| GS-Grouping [67] | 32.4 | 69.5 | 50.0 | 44.6 | 49.1 | 26.4 | 54.0 | 31.3 | 34.6 | 36.6 |
| LEGaussian [49] | 69.0 | 79.7 | 63.6 | 57.1 | 67.4 | 20.2 | 32.3 | 22.3 | 23.4 | 24.6 |
| GOI [52] | 56.3 | 67.8 | 68.2 | 44.6 | 59.2 | 33.7 | 55.8 | 54.5 | 23.9 | 42.0 |
| GAGS [15] | 69.0 | 88.1 | 90.9 | 78.6 | 81.7 | 46.8 | 60.3 | 55.8 | 53.6 | 54.1 |
| LangSplat [12] | 73.2 | 88.1 | 95.5 | 80.4 | 84.3 | 51.2 | 65.1 | 44.5 | 44.7 | 51.4 |
| LangSplatV2 | 74.7 | 93.2 | 86.4 | 82.1 | 84.1 | 51.8 | 72.2 | 59.1 | 56.4 | 59.9 |

Specifically, each Gaussian's $L$-dimensional sparse coefficient $\boldsymbol{w}_i$ can be fully represented by its top-$K$ non-zero elements. We store these as two $K$-dimensional arrays for each Gaussian: top-$K$ indices and top-$K$ coefficients. The top-$K$ indices are the positions of the top-$K$ non-zero elements within the $L$-dimensional vector and the top-$K$ coefficients are the values of these top-$K$ non-zero elements. During rendering, each CUDA thread performs alpha-blending only for the $K$ non-zero coefficients. For each Gaussian point within the tile, the CUDA thread will access the indices and coefficients of the top-$K$ elements and perform weighted summation solely on these $K$-dimensional indices, avoiding computation on zero elements. In this way, LangSplatV2 can obtain high-dimensional ($D$) feature splatting results at the cost of only ultra-low-dimensional ($K$) feature splatting.

## 4 Experiments

### 4.1 Datasets and Details

**Datasets.** We evaluate our method on the LERF, 3D-OVS, and Mip-NeRF360 datasets. The LERF dataset [13], captured using the iPhone App Polycam, contains in-the-wild scenes. For the open-vocabulary 3D object localization task, we adopt the augmented localization annotations provided by LangSplat [12] on the LERF dataset. Additionally, we use the segmentation ground truth from LangSplat [12] for the open-vocabulary 3D segmentation task on LERF. Beyond LERF, we also conduct 3D segmentation experiments on the 3D-OVS and Mip-NeRF360 [66] datasets. The 3D-OVS dataset [47] includes a collection of long-tail objects captured in diverse poses and backgrounds. Moreover, we evaluate our method on Mip-NeRF360, which consists of multi-view indoor and outdoor scene images, with segmentation labels annotated by GAGS [15]. For evaluation, we use localization accuracy for the 3D object localization task and report the average IoU scores for the open-vocabulary 3D segmentation task.

**Implementation Details.** Following LangSplat [12], we use the OpenCLIP ViT-B/16 model to extract CLIP features. We employ the ViT-H model for SAM [65] to segment images and obtain masks with three hierarchical semantics. The codebook size $L$ is set to 64 and the $K$ is set to 4. During test-time querying, we render three semantic scales simultaneously, leading to the actual rendering dimension of 12. The 3D Gaussians are first trained with RGB supervision for 30,000 iterations to reconstruct the RGB scene. Then we train another 10,000 iterations for the 3D sparse coefficient field by fixing all other 3D Gaussian parameters. All our experiments are conducted on one A100 GPU.

### 4.2 Quantitative Results

**Time Analysis.** To assess the speed improvement of LangSplatV2 over its LangSplat, we conducted a detailed time analysis using the LERF dataset, with the computations performed on a single A100 GPU. Table 1 presents a stage-wise breakdown of the time costs for LangSplat, LangSplat*, and LangSplatV2. LangSplat, our initial model, achieves a frame rate of 8.2 FPS. The LangSplat* improvements mainly focused on rendering and post-processing optimizations, which reduced the total time to 85.6 ms and increased the speed to 11.7 FPS. Here, the rendering time was significantly cut to 2.0 ms, and the post-processing time was reduced to less than 1.0 ms with our proposed simple engineering modifications. In contrast, LangSplatV2 proposed an efficient sparse coefficient splatting method, that entirely removed the MLP decoder in the decoding stage. The only operation in the

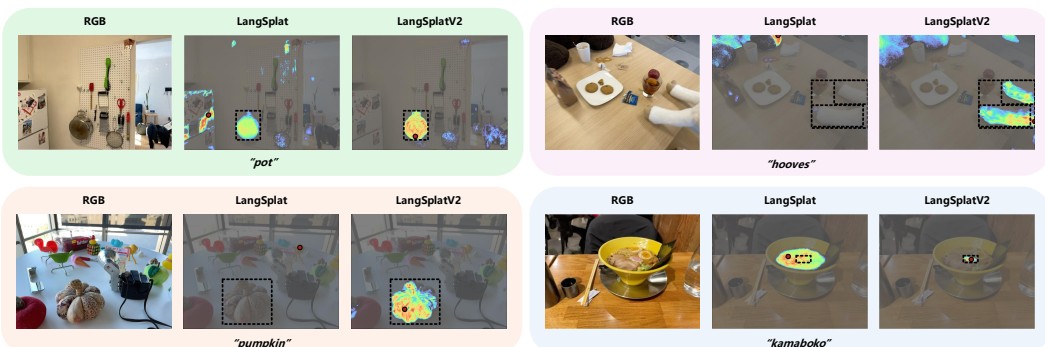

Figure 4: Qualitative comparisons of open-vocabulary 3D object localization on the LERF dataset. The red points are the model predictions and the black dashed bounding boxes denote the annotations. We observe that LangSplatV2 generates better results than LangSplat.

decoding stage is performing a matrix multiplication between the rendered $L$-dimensional coefficient and the global dictionary, which takes only 0.1 ms. While completely removing the MLP decoder, our LangSpaltV2 only needs 2.0 ms to render the sparse coefficient, leading to an overall 476.2 FPS for obtaining high-dimensional feature maps. In the end, LangSplatV2 achieved an exceptional reduction in total querying time to 2.6 ms. These results not only demonstrate a substantial enhancement in processing speed but also affirm the real-time capabilities of LangSplatV2 for high-resolution, complex 3D scene querying.

**Main Results on the LERF dataset.**
Table 2 demonstrates the comparisons on the LERF dataset. For the open-vocabulary 3D object localization task, LangSplatV2 achieved the highest accuracy with notable improvements over previous methods for most scenes, including "ramen", "figurines", and "teatime". In the "Kitchen" scene, LangSplat achieved a higher accuracy than LangSplatV2. This difference is mainly due to the limited sample size (22 examples in total), with only a two-prediction gap (21 vs. 19). Given LangSplat's high

Table 3: Quantitative 3D semantic segmentation results on 3D-OVS, reported as mean IoU (%).

| Method | Bed | Bench | Room | Sofa | Lawn | Overall |
|---|---|---|---|---|---|---|
| FFD [1] | 56.6 | 6.1 | 25.1 | 3.7 | 42.9 | 26.9 |
| LERF [13] | 73.5 | 53.2 | 46.6 | 27.0 | 73.7 | 54.8 |
| 3D-OVS [47] | 89.5 | 89.3 | 92.8 | 74.0 | 88.2 | 86.8 |
| GS-Grouping [67] | 83.0 | 91.5 | 85.9 | 87.3 | 90.6 | 87.7 |
| LEGaussian [49] | 84.9 | 91.1 | 86.0 | 87.8 | 92.5 | 88.5 |
| GOI [52] | 89.4 | 92.8 | 91.3 | 85.6 | 94.1 | 90.6 |
| LangSplat [12] | 92.5 | 94.2 | 94.1 | 90.0 | 96.1 | 93.4 |
| LangSplatV2 | **93.0** | **94.9** | **96.1** | **92.3** | **96.6** | **94.6** |

variance in this scene, we ran it multiple times, obtaining an average of $17.5 \pm 1.8$ correct predictions, which suggests that LangSplatV2 provides more consistent performance advantage across different scenarios. Results on 3D semantic segmentation demonstrate that LangSplatV2 outperforms all other methods, and it consistently surpasses LangSplat across all scenes, highlighting the effectiveness of our proposed method.

**Main Results on the 3D-OVS dataset.** Table 3 shows the quantitative comparisons of open-vocabulary 3D semantic segmentation performance, on the 3D-OVS dataset across five test scenes. We see that LangSplatV2 achieves the highest overall mean IoU score of 94.6%. It significantly outperforms LangSplat across all different scenes, further validating its ability to build more accurate language fields.

Table 4: Quantitative 3D semantic segmentation results on Mip-NeRF360, reported as mean IoU (%).

| Method | Room | Counter | Garden | Bonsai | Overall |
|---|---|---|---|---|---|
| GS-Grouping [67] | 54.4 | 47.7 | 40.4 | 54.1 | 49.2 |
| LEGaussian [49] | 25.5 | 35.3 | 33.2 | 22.3 | 29.1 |
| GOI [52] | 60.3 | 46.6 | 59.8 | 67.3 | 58.5 |
| GAGS [15] | **65.2** | 61.1 | 61.2 | 70.5 | 64.5 |
| LangSplat [12] | 53.2 | 68.8 | 51.9 | 55.4 | 57.3 |
| LangSplatV2 | 64.3 | **75.1** | **65.0** | **73.1** | **69.4** |

**Main Results on the Mip-NeRF360 dataset.** Table 4 presents the comparison results on the Mip-NeRF360 dataset. Notably, LangSplatV2 achieved a significant improvement in the overall average IoU score, reaching 69.4%, which is a marked advancement over LangSplat's 57.3%.

Table 5: Ablation study on codebook size $L$. We report the average performance of the 3D object localization and 3D semantic segmentation tasks on the LERF dataset.

| $L$ | 32 | 64 | 128 |
|---|---|---|---|
| Accuracy (%) | 72.8 | 84.1 | 84.1 |
| IoU (%) | 53.9 | 59.9 | 60.5 |

Table 6: Ablation study on different $K$. We report the average performance of the 3D object localization and 3D semantic segmentation tasks on the LERF dataset.

| $K$ | 2 | 4 | 6 | 8 |
|---|---|---|---|---|
| Accuracy (%) | 79.4 | 84.1 | 84.4 | 83.6 |
| IoU (%) | 54.4 | 59.9 | 59.9 | 59.9 |

Figure 5: Qualitative comparisons of open-vocabulary 3D semantic segmentation on the LERF, Mip-NeRF360 and 3D-OVS dataset. We can see that our LangSplatV2 generates better masks than LangSplat, which shows the effectiveness of our LangSplatV2.

**Ablation Study.** In Table 5, we evaluate the effect of codebook sizes $L$ on the localization and segmentation tasks. We report the mean accuracy and IoU scores on the LERF dataset. Our results reveal that increasing L from 32 to 64 yields a notable improvement in both localization accuracy (72.8% to 84.1%) and segmentation IoU (53.9% to 59.9%). This suggests that a larger codebook size better captures the semantic fields in complex scenes, allowing for more precise language representations. However, further increasing $L$ results in only slightly increased performance, suggesting that $L = 64$ has already reached saturation on the LERF dataset. Table 6 further reports the effects of different $K$. We observe that $K = 4$ can saturate the performance of the Gaussian model, and a larger K will increase the rendering dimensions, thereby slowing down the rendering speed. More ablation studies and details can be found in the supplementary material.

**Training Cost.** Table 7 shows the comparison of training cost on the LERF [13] dataset with one A100 GPU. Compared with LEGaussians [49] and LangSplat [12], LangSplatV2 comes with increased training cost due to the need to construct high-dimensional semantic fields during training. Although our LangSplatV2 incurs higher training costs compared to LangSplat and LEGaussians, our primary focus in this paper is on improving the model's test-time performance for deployment. As demonstrated in the experiments, LangSplatV2 achieves significantly higher inference speed—47× faster than LangSplat and 14× faster than LEGaussians on the LERF dataset—under comparable memory consumption, while also yielding higher segmentation accuracy. Furthermore, compared with naive LangSplat (directly trained with 512 dimensions), LangSplatV2 is more efficient with the proposed global codebook and sparse coefficient field.

**Discussion.** Recent work has also explored codebook-based language field representations, such as LEGaussians [49]. However, our approach departs significantly from theirs in several key aspects.

Table 7: Quantitative comparisons of training cost on the LERF dataset.

| Method | LangSplat [12] | LEGaussian [49] | LangSplatV2 | LangSplat-512D [12] |
|---|---|---|---|---|
| **Training Time (h)** | 1.0 | 1.3 | 3.0 | 45.0 |
| **Training Memory (GB)** | 6.2 | 11 | 21.2 | 47.4 |

Table 8: Quantitative comparisons with LEGaussian on three benchmarks.

| Method | Segmentation IoU (%) | | | Query Time (ms) | | | GPU Memory (GB) | | |
|---|---|---|---|---|---|---|---|---|---|
| | LERF | 3DOVS | Mip. | LERF | 3DOVS | Mip. | LERF | 3DOVS | Mip. |
| LEGaussian [49] | 24.6 | 88.5 | 29.1 | 36.7 | 59.6 | 58.0 | 8.2 | **10.5** | **14.0** |
| LangSplatV2 | **59.9** | **94.6** | **69.4** | **2.6** | **4.8** | **3.8** | **7.2** | 11.6 | 16.9 |

First, LEGaussians construct a 2D feature codebook from 2D images while LangSplatV2 directly learns a global codebook in 3D space, which is more efficient. Second, LEGaussians still rely on an MLP to compress features and do not exploit sparsity. In contrast, our method completely removes the MLP and further enforces and leverages sparsity in both representation and rendering. As shown in Table 8, our method achieves superior semantic segmentation accuracy, faster inference speed, and comparable memory usage across three benchmark datasets, which demonstrates the effectiveness of our approach. For a more detailed comparison and discussion, please refer to the Appendix.

## 4.3 Qualitative Results

Figure 4 visualizes the open-vocabulary 3D object localization results. We observe that our LangSpaltV2 can give more accurate localization predictions compared with LangSplat. For example, our LangSplatV2 can give the correct prediction for "pumpkin" while LangSplat entirely fails in this query. Figure 5 visualizes the open-vocabulary 3D segmentation results in three datasets. We observe that our LangSplatV2 can generate more accurate masks compared with LangSplat. For example, in the "Kitchen" scene, LangSplat predicts a noisy mask for the "ketchup" query, while our LangSplatV2 generates more precise and clean masks.

## 5 Conclusion

**Conclusion.** In this paper, we have presented LangSplatV2 for high-dimensional language feature splatting in 3D open-vocabulary querying, designed to overcome the speed limitations of LangSplat. Our analysis identified the decoding stage as the primary bottleneck in LangSplat. By modeling each Gaussian point as a sparse code over a global dictionary, LangSplatV2 removes the need for a heavyweight decoder. We further proposed an efficient sparse coefficient splatting method with CUDA optimization, allowing LangSplatV2 to achieve high-dimensional feature splatting results at the cost of splatting ultra-low-dimensional features.

**Limitations and broader impacts.** While LangSplatV2 demonstrates improved performance and faster inference compared to LangSplat, it comes with increased training cost due to the need to construct high-dimensional semantic fields during training. Furthermore, as our method directly inherits the semantic representations from the CLIP model, it also inherits its inherent biases. Addressing such biases remains an open research challenge and may benefit from recent advances in fairness-aware versions of CLIP.

## Acknowledgment

This research is supported in part by the NIH grant R01HD104969.

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

# A Algorithm

The proposed efficient sparse coefficient splatting process is shown in Algorithm 1.

---

**Algorithm 1** Efficient Sparse Coefficient Splatting

---

Initialize $\boldsymbol{W} \in \mathbb{R}^L$ to zero
**for** Gaussian point $i \in \mathcal{N}$ **do**
    Retrieve top-$K$ indices $\{j_1, j_2, ..., j_K\}$ and corresponding coefficients $\{w_{i,j_1}, w_{i,j_2}, ..., w_{i,j_K}\}$
    **for** each $j \in \{j_1, ..., j_K\}$ **do**
        $\boldsymbol{W}[j] \mathrel{+}= w_{i,j} \cdot \alpha_i \cdot \prod_{m=1}^{i-1}(1 - \alpha_m)$
    **end for**
**end for**

---

# B Discussion

LangSplatV2 proposes to model the high-dimensional language features of each Gaussian point as a sparse code within a global dictionary. While similar concepts, such as quantizing Gaussian point attributes (*e.g.*, SH coefficients [56], opacity [53], and semantic features [49]) into a codebook, have been explored in the domains of compression [53] and language embedding [49], our method is fundamentally different.

In the compression domain, existing approaches [53, 56] first learn unique attributes for each Gaussian point, which are then quantized into a codebook using a quantizer. Extending this concept to our scenario would require learning 512-dimensional semantic features for each Gaussian point before quantization. This process significantly increases memory consumption and training time, often exceeding the capacity of GPUs like the NVIDIA 3090 or 4090, leading to out-of-memory (OOM) errors. Moreover, even with advanced GPUs that can handle the training, the rendering process would still involve managing 512-dimensional features. As shown in Figure 1, this dramatically reduces rendering speed compared to LangSplatV2.

In the context of language embedding, LEGaussians [49] adopt a codebook-based approach to accelerate training. However, our method is significantly different from LEGaussians in three key aspects: 1) Global 3D Codebook vs. 2D Codebook: LEGaussians first train a 2D codebook by quantizing features in 2D images, then learn a 3D model to predict the one-hot class category indicating the index of the codebook. In contrast, LangSplatV2 learns a global 3D codebook shared among all 3D Gaussian points. By avoiding the additional reconstruction errors caused by first quantizing features in the 2D image plane, our approach reduces overall reconstruction error and more efficiently utilizes the codebook in 3D space. 2) Eliminating the MLP Bottleneck: LEGaussians rely on an MLP to project each Gaussian point's features into a one-hot class category representing the index of the codebook. Our method removes the need for an MLP entirely, thereby eliminating the speed bottleneck and improving computational efficiency. 3) Efficient Sparse Coefficient Splatting: LangSplatV2 introduces Efficient Sparse Coefficient Splatting, which reduces the effective rendering dimensions to $K$. In contrast, LEGaussians renders with the full feature dimensions of Gaussian points, which inherently slows down the rendering process.

Furthermore, due to the reliance on an MLP decoder, LEGaussians still operate in a lower-dimensional feature space, which limits the expressiveness of its learned representations. Additionally, its two-stage 2D codebook approach fails to effectively incorporate 3D priors, resulting in a suboptimal learned field. In contrast, LangSplatV2 directly models high-dimensional features within a global 3D codebook, effectively capturing the underlying 3D structures. As evidenced by the results in Table 8, our approach outperforms LEGaussians across multiple benchmarks while being significantly faster, demonstrating superior scene reconstruction quality and more accurate language-grounded 3D representations.

To conclude, LangSplatV2 demonstrates a significant leap forward in the efficiency and scalability of modeling high-dimensional language features in 3D scenes. By leveraging the proposed 3D sparse coefficient fields and efficient sparse coefficient splatting techniques, our method reduces computational overhead while maintaining high fidelity in 3D language scene rendering.

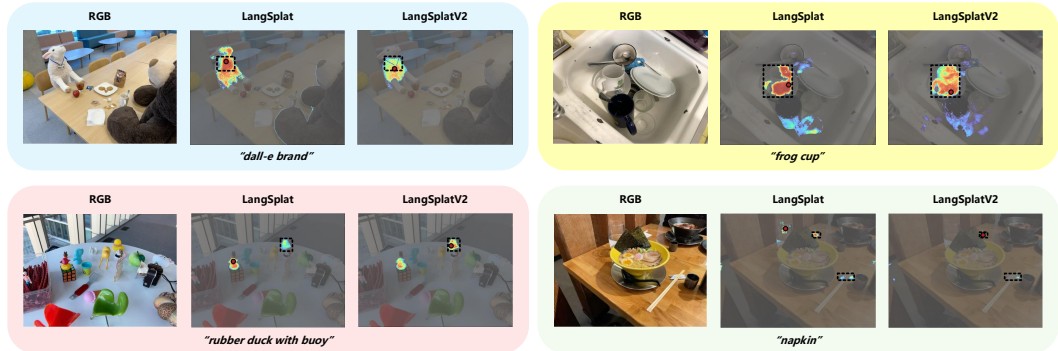

Figure 6: More qualitative comparisons of open-vocabulary 3D object localization on the LERF and Mip-NeRF360 datasets. The red points are the model predictions and the black dashed bounding boxes denote the annotations. We observe that LangSplatV2 generates better results than LangSplat.

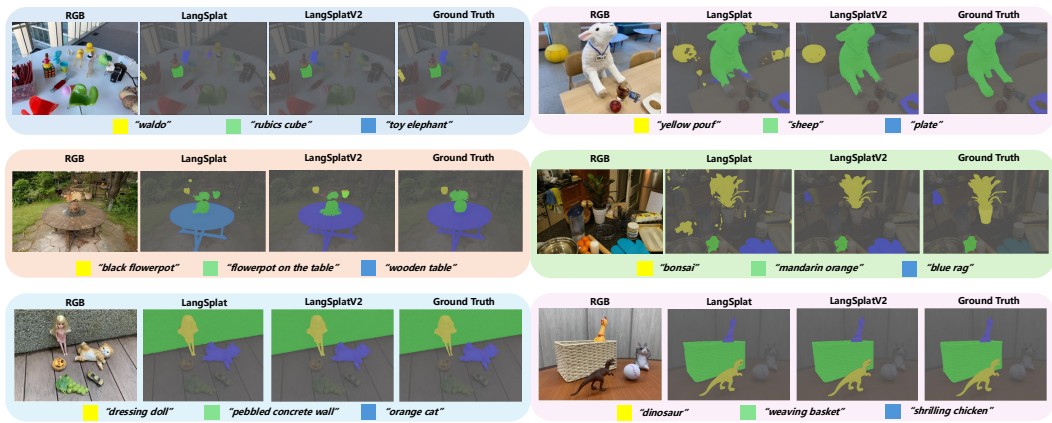

Figure 7: More qualitative comparisons of open-vocabulary 3D semantic segmentation on the LERF, Mip-NeRF360 and 3D-OVS dataset. We can see that our LangSplatV2 generates better masks than LangSplat, which shows the effectiveness of our LangSplatV2.

## C  Evaluation Details

After calculating the similarity between the rendered semantic features and the CLIP features of the query text, we obtained relevance maps $M$ at different semantic levels. Then, we use different strategies for different datasets to obtain the corresponding query results.

**LERF.** For object localization, we first apply average pooling to smooth $M$ and get $Avg(M)$. Then we select the point with the highest relevancy in $Avg(M)$ as the predicted point. Additionally, we choose the semantic level corresponding to the highest relevancy value among the three levels as the selected level. For semantic segmentation, after getting $Avg(M)$, we select the level in the same way as object localization. We then normalize $Avg(M)$ and use a threshold to obtain a binary mask as the final prediction mask.

**3D-OVS.** The post-processing pipeline for semantic segmentation is similar to that used for the LERF dataset. Differently, we obtain the smoothed relevancy map $Avg'(M)$ as $Avg'(M) = 0.5 \times (M + Avg(M))$, and the level selection is based on the average relevancy within the predicted mask. Additionally, when evaluating query accuracy, we use two coarse-grained semantic levels (*whole* and *part*) owing to the simplicity of the scenes in this dataset.

**Mip-NeRF360.** Similar to the post-processing for the 3D-OVS dataset, we obtain the smoothed relevancy map $Avg'(M)$ and select the semantic level based on the average relevancy score. When

evaluating query accuracy, we use all three semantic levels, as the complex scenes in this dataset demand finer-grained language feature rendering.

## D  More Ablation Study

**Time Analysis.** We conduct ablation studies on the LERF [13] dataset and report the render and decode speed (ms per query) in Table 9. We test the speed on one A100 GPU. The baseline is LangSplat, which renders three-level 3-dimensional language features separately. Rendering three semantic levels in parallel can reduce the rendering time from 6.0 ms to 2.0 ms per query. The sparse coefficient field can significantly speed up the decoding stage from 83.1 ms/q to 0.1 ms/q by changing the heavy weight decoder to a simple matrix multiplication operation. Efficient sparse coefficient splatting method effectively transformed the 192-dimensional rendering into 12-dimensional rendering, reducing the rendering time from 5.3 ms to 2.0 ms per query through CUDA optimization.

Table 9: Speed ablation study on the LERF dataset. Parallel means rendering three semantic levels in parallel, Sparse means 3D sparse coefficient field, Efficient means efficient sparse coefficient splatting with CUDA optimization.

| Component | | | Speed (ms/q) | | |
|---|---|---|---|---|---|
| Parallel | Sparse | Efficient | render | decode | total |
| | | | 6.0 | 83.1 | 89.1 |
| ✓ | | | 2.0 | 83.1 | 85.1 |
| ✓ | ✓ | | 5.3 | 0.1 | 5.4 |
| ✓ | ✓ | ✓ | **2.0** | **0.1** | **2.1** |

$L$ **and** $K$**.** In Table 10 and Table 11, we show the ablation study results on each scene in the LERF [13] dataset. As we can see, for some complex scenarios, such as Kitchen and Figurines, larger $L$ and $K$ can lead to better performance. However, for all scenarios, the model's performance is already good enough with the settings of $L = 64$ and $K = 4$, and increasing $L$ and $K$ will lead to increased computational resources and time cost for training and inference. Therefore, we set $L = 64$ and $K = 4$ to balance performance and efficiency.

Table 10: Ablation study on codebbok size $L$. We report the localization and the segmentation performance on the LERF dataset.

| | $L$ | 32 | 64 | 128 |
|---|---|---|---|---|
| Ramen | Accuracy (%) | 70.4 | **74.7** | 74.7 |
| | IoU (%) | 50.5 | **51.8** | 51.4 |
| Teatime | Accuracy (%) | 84.8 | **93.2** | 91.5 |
| | IoU (%) | 65.7 | **72.2** | 69.8 |
| Kitchen | Accuracy (%) | 68.2 | 86.4 | **86.4** |
| | IoU (%) | 54.2 | 59.1 | **63.2** |
| Figurines | Accuracy (%) | 67.9 | 82.1 | **83.9** |
| | IoU (%) | 45.3 | 56.4 | **57.6** |
| Overall | Accuracy (%) | 72.8 | **84.1** | 84.1 |
| | IoU (%) | 53.9 | 59.9 | **60.5** |

## E  More Visualization Results

**Open-vocabulary 3D Object Localization.** We visualize more examples on the LERF [13] and Mip-NeRF360 [13] datasets for open-vocabulary 3D object localization in Figure 6.

Table 11: Ablation study on different $K$. We report the localization and the segmentation performance on the LERF dataset.

|  | $K$ | 2 | 4 | 6 | 8 |
|---|---|---|---|---|---|
| Ramen | Accuracy (%) | 71.8 | **74.7** | 71.8 | 73.2 |
|  | IoU (%) | 50.2 | **51.8** | 51.4 | 51.1 |
| Teatime | Accuracy (%) | 91.5 | **93.2** | 91.5 | 91.5 |
|  | IoU (%) | 68.8 | **72.2** | 70.7 | 70.6 |
| Kitchen | Accuracy (%) | 77.3 | 86.4 | **95.5** | 90.9 |
|  | IoU (%) | 48.9 | 59.1 | 59.8 | **60.1** |
| Figurines | Accuracy (%) | 76.8 | **82.1** | 78.6 | 78.6 |
|  | IoU (%) | 49.6 | 56.4 | **57.7** | 57.6 |
| Overall | Accuracy (%) | 79.4 | 84.1 | **84.4** | 83.6 |
|  | IoU (%) | 54.4 | **59.9** | 59.9 | 59.9 |

**Open-vocabulary 3D Semantic Segmentation.** We visualize more examples on the LERF [13], Mip-NeRF360 [66] and 3D-OVS [47] datasets for open-vocabulary 3D semantic segmentation in Figure 7.

