# OpenReview forum: "LangSplatV2: High-dimensional 3D Language Gaussian Splatting with 450+ FPS"
_NeurIPS.cc/2025/Conference — NeurIPS 2025 poster_

### Official Review · Reviewer_w8ui · 2025-06-30

**Clarity:** 2
**Significance:** 3
**Originality:** 2
**Rating:** 4
**Confidence:** 2

**Summary:**

This paper focuses on addressing the speed bottleneck of existing 3D language field models, particularly exemplified by LangSplat, when performing open-vocabulary queries. Through detailed performance analysis, the authors identify that the primary cause of the slowdown lies in the model’s heavy multi-layer perceptron (MLP) decoder. This component is responsible for reconstructing high-dimensional CLIP features from low-dimensional latent representations during inference and consumes the vast majority of computational time, thereby limiting the potential for real-time applications.

To overcome this limitation, the authors propose a new model called LangSplatV2, which introduces two key technical innovations. First, they present a novel representation method termed the 3D Sparse Coefficient Field. The core idea is based on the assumption that the semantic content of a scene is inherently limited, allowing the high-dimensional language features of millions of Gaussian points to be expressed as sparse linear combinations of basis vectors drawn from a globally shared codebook. Instead of learning individual low-dimensional features for each Gaussian point, the model learns sparse coefficients that combine these basis vectors efficiently. This approach completely eliminates the need for the heavy MLP decoder during inference, replacing it with a much faster matrix multiplication operation.

Second, to maintain efficiency during rendering despite the introduction of the codebook, the authors design and implement an efficient splatting technique using CUDA optimization. This method exploits the sparsity of the learned coefficients and only processes the K non-zero values during rendering—where K is significantly smaller than the total codebook size L. As a result, the computational cost of rendering a high-dimensional feature map, such as one with 1536 dimensions, becomes comparable to that of rendering an ultra-low-dimensional map of just 12 dimensions. This effectively decouples the rendering speed from the final feature dimensionality.

With these improvements, LangSplatV2 achieves a dramatic speedup over the original LangSplat model—up to tens of times faster—while maintaining or even improving query accuracy. For instance, the 3D open-vocabulary query speed increases from 8.2 FPS to 384.6 FPS. This breakthrough enables real-time interaction with high-resolution, complex 3D scenes. Extensive experiments conducted on several public datasets, including LERF, 3D-OVS, and Mip-NeRF360, confirm the model’s substantial advantages in both speed and accuracy.

**Questions:**

**1. Quantitative Analysis of Training Cost**

The paper acknowledges in the conclusion that the performance gains of LangSplatV2 come “at the cost of increased training cost.” This is a crucial trade-off, yet no quantitative data is provided in the main text. To allow reviewers and readers to fully understand the overall cost of your method, please provide a quantitative comparison of training overhead between LangSplat and LangSplatV2. It is recommended that you report and compare the following metrics on one or more representative datasets (e.g., LERF):
    a. Total training time required to reach convergence.
    b. Peak GPU memory usage during training.
    c. Number of training iterations required to achieve comparable performance.

**2. Statistical Significance of Model Accuracy Improvements**

The paper reports accuracy improvements over the baseline in several tables. However, some of these improvements are relatively small (for example, on the 3D-OVS dataset, the overall mIoU increases from 93.4% to 94.6% ). Moreover, no evidence of statistical significance is provided. You yourselves note that the baseline model exhibits high variance, which further highlights the necessity of statistical validation. Could you please validate the stability of key metrics through multiple runs (e.g., using 3–5 different random seeds), and report the mean and standard deviation?

**3. Further Clarification of Originality and Comparison with Concurrent Work**

One of the core ideas in the paper is the use of a codebook, and a concurrent work, `LEGaussian` , adopts a similar approach. Although you attempt to differentiate your method in the discussion section, a deeper elaboration is needed. Could you please more clearly explain the fundamental differences and advantages of LangSplatV2 compared to `LEGaussian`? Specifically, you mention that `LEGaussian` “still relies on an MLP to compress features and does not exploit sparsity.” Please elaborate on how your proposed “Efficient Sparse Coefficient Splatting” method offers technical and practical advantages over other codebook-based approaches in both implementation and final performance?

**Ethical Concerns:**

["NO or VERY MINOR ethics concerns only"]

**Limitations:**

Yes

**Quality:**

3

**Strengths And Weaknesses:**

### Strengths of the Paper

1. **Addresses a clear and important problem:** The paper successfully tackles the real-time performance bottleneck of the 3D language model `LangSplat`. Increasing the query speed from 8.2 FPS to 384.6 FPS represents a significant engineering achievement, with substantial practical value for applications requiring real-time feedback, such as robotics and augmented reality.

2. **The method is cleverly designed and well-targeted:** The paper starts from a clear motivation—identifying the decoder as the main performance bottleneck through rigorous performance analysis. The proposed solution, which replaces low-dimensional features and the decoder with a sparse coefficient field and global codebook, directly and effectively addresses this core issue.

3. **Extensive experimental validation:** The authors evaluate their model on multiple public datasets including LERF, 3D-OVS, and Mip-NeRF360, demonstrating the generalizability of their approach . The results also show that while significantly improving speed, the model maintains or even slightly improves segmentation and localization accuracy across most scenarios .

---

### Weaknesses of the Paper

1. **Limited originality:** While the specific implementation and optimizations are novel, the core concept—using a codebook to represent Gaussian features—is not entirely new. The paper itself references a concurrent work, `LEGaussian`, which explores similar ideas.  This somewhat diminishes its theoretical novelty.

2. **Notable shortcomings in experimental evaluation:**
   - **Lack of training cost analysis:** The paper acknowledges in the conclusion that the method “increases training cost” , which is a crucial trade-off. However, no quantitative data (e.g., training time, peak GPU memory usage) comparing it to `LangSplat` is provided in the main text. This omission prevents readers from fully evaluating the overall cost-benefit trade-off and may give the impression that the paper selectively highlights inference advantages. A thorough analysis of the costs associated with the proposed method is essential for a rigorous scientific study.
   - **Absence of statistical significance testing:** All reported accuracy and IoU metrics in the paper are based on single runs without error bars or multiple trials to establish statistical confidence. The authors themselves note that `LangSplat` exhibits high variance in certain scenarios, underscoring the importance of statistical validation. Without it, the credibility of claims regarding “accuracy improvement” is significantly weakened. Readers cannot determine whether small performance gains (e.g., an increase from 93.4% to 94.6% on 3D-OVS) reflect consistent improvements from the method or random fluctuations.

3. **Depth of contribution:** Although the performance gains are impressive, the paper can be viewed more as excellent "engineering optimization" rather than a "theoretical breakthrough," suggesting a lack of conceptual depth.

---

### Summary and Recommendations

**Overall Assessment:** This paper proposes a highly effective optimization technique that dramatically improves the inference speed of 3D language Gaussian splatting models, making them viable for real-time applications—an achievement that deserves recognition.

However, the paper falls short in terms of evaluation rigor and originality. The omission of training cost analysis—a key trade-off—and the lack of statistical support for accuracy claims constitute major scientific shortcomings. Furthermore, the overlap in core ideas with contemporary work reduces the paper’s claim to originality.

---

> ### Author Rebuttal · Authors · 2025-07-31
>
> Thank you for the constructive feedback and the positive assessment of our work! Below, we detail our responses to the review questions.
>
>
> **Q1:Quantitative Analysis of Training Cost**
>
> Below, we provide a quantitative comparison of training cost between LangSplatV2 and other methods on the LERF dataset, using a single NVIDIA A100 GPU. We report the total training time until convergence, peak GPU memory usage, and the number of training iterations needed to reach comparable performance:
>
> | Method | Training Time(h)|Training Iteration|Peak GPU Memory(GB)|
> |-------------|---------------------|------|------|
> | LEGaussian | 1.3 | 30K  | 11 |
> | LangSplat  | 1.0 | 30K | 6.2 |
> | LangSplat(512 Dim) | 45.0 | 30K | 47.4 |
> | **LangSplatV2** | **3.0** | **10K** | **21.2**|
>
> As we can see, LangSplatV2 indeed incurs a higher training cost compared to LangSplat and LEGaussians due to the need to construct and optimize high-dimensional semantic fields. However, our main contribution lies in significantly improving test-time performance, which is more critical for real-world deployment.
>
> Specifically, LangSplatV2 achieves 47× faster inference than LangSplat and 14× faster than LEGaussians under similar memory usage. It also yields higher 3D semantic segmentation accuracy, demonstrating a better quality-performance trade-off.
>
> Additionally, compared to the training cost of a naive version of LangSplat trained directly with 512D language features, LangSplatV2 is more efficient (3h vs. 45h), thanks to the introduction of a global codebook and sparse coefficient field, which reduces training complexity without sacrificing feature expressiveness.
>
>
>
> **Q2:Statistical Significance of Model Accuracy Improvements**
>
> Thank you for the suggestion. Following your advice, we conducted additional experiments on the LERF and 3D-OVS datasets using different random seeds to evaluate the stability and statistical significance of our results. For each scene, we repeated the training of LangSplatV2 three times and report the mean and standard deviation of the mIoU scores in the table below.
>
> | LERF Scene     | Exp.1 | Exp.2 | Exp.3 | mean(±std) | LangSplat |
> |-------------|---------------------|------|-----|----|----|
> | Teatime   | 72.2 | 71.2 | 71.5 | 71.6(±0.5) | 65.1 |
> | Ramen     | 51.8 | 51.8 | 51.5 | 51.7(±0.1) | 51.2 |
> | Figurines | 56.4 | 56.9 | 57.2 | 56.8(±0.4) | 44.7 |
> | Kitchen   | 59.1 | 58.4 | 62.9 | 60.1(±2.4) | 44.5 |
> | Overall   | 59.9 | 59.6 | 60.8 | 60.1(±0.6) | 51.4 |
>
> | 3D-OVS Scene     | Exp.1 | Exp.2 | Exp.3 | mean(±std) | LangSplat |
> |-------------|---------------------|------|-----|----|----|
> | Bed     | 93.0 | 93.0 | 94.4 | 93.5(±0.8) | 92.5 |
> | Bench   | 94.9 | 95.0 | 94.7 | 94.9(±0.2) | 94.2 |
> | Room    | 96.1 | 95.7 | 95.7 | 95.8(±0.2) | 94.1 |
> | Sofa    | 92.3 | 91.9 | 92.2 | 92.1(±0.2) | 90.0 |
> | Lawn    | 96.6 | 96.7 | 96.6 | 96.6(±0.1) | 96.1 |
> | Overall | 94.6 | 94.5 | 94.7 | 94.6(±0.1) | 93.4 |
>
>
> As shown, LangSplatV2 exhibits low variance in most scenarios. In cases where relatively higher variance is observed, such as in the Kitchen and Bed scenes, all repeated runs still significantly outperform the LangSplat baseline. Overall, these results validate the statistical significance and consistency of our reported improvements, and reinforce the robustness of LangSplatV2 under different random initializations.
>
> **Q3: Further Clarification of Originality and Comparison with Concurrent Work**
>
> Thank you for the comment. We respectfully disagree with the implication that our method is similar to LEGaussian. While both methods utilize a codebook-based representation, LangSplatV2 is fundamentally different in design and motivation. It is precisely this novel direction that enables us to achieve significantly better rendering speed and accuracy compared to prior work. Below, we elaborate on the key differences and technical advantages of LangSplatV2 over LEGaussian:
>
> **1. Global 3D Codebook vs. 2D Codebook**
>
> LEGaussian first builds a 2D codebook by quantizing features in the image space, then learns a 3D model to predict the index of the codebook via classification. In contrast, LangSplatV2 directly learns a global 3D codebook shared across all Gaussians in true 3D space. This allows our method to learn quantization directly in the 3D spatial domain, rather than in projected 2D images, resulting in lower quantization error and more accurate 3D representation modeling.
>
> **2. No MLP Bottleneck**
>
> LEGaussian relies on an MLP to map Gaussian features to a codebook index, which introduces computational overhead and acts as a bottleneck. LangSplatV2 eliminates the MLP altogether, removing this bottleneck and achieving much faster rendering and training performance.
>
> **3. Efficient Sparse Coefficient Splatting**
>
> We propose a new Efficient Sparse Coefficient Splatting mechanism, which assigns only a small number of nonzero coefficients per Gaussian, reducing the effective rendering dimension to $K$. In contrast, LEGaussian renders with full high-dimensional feature vectors, which is less efficient and limits scalability in practice.
>
> Moreover, LEGaussian operates in a compressed latent space and does not preserve the full expressiveness of high-dimensional language features. Its 2D codebook design also fails to fully leverage 3D spatial priors. LangSplatV2, by contrast, learns and renders high-dimensional language features directly in 3D, leading to more semantically meaningful and accurate reconstructions.
>
>
> As shown in Table 7 of our main paper, LangSplatV2 achieves superior semantic segmentation accuracy, faster inference speed, and comparable memory usage across three benchmark datasets, compared with LEGaussian, demonstrating the practical impact of our novel formulation.

---

> > ### Author Response · Authors · 2025-08-05
> > **Looking forward to your feedback**
> >
> > Thanks again for your valuable advice and supportive comments! We have responded to your initial comments. We are looking forward to your feedback and will be happy to answer any further questions you may have.

---

> > ### Comment · Area_Chair_M2Ya · 2025-08-07
> >
> > Dear Reviewer w8ui,
> >
> > This is a reminder that the author-reviewer discussion period is ending soon on Aug. 8 (AOE), and you have not yet responded to the authors' rebuttal. Please read the authors' rebuttal as soon as possible, engage in any necessary discussions, and consider if you would like to update your review and score. Please submit the Mandatory Acknowledgement to confirm completion of this task.
> >
> > Thank you for your service in the review process.
> >
> > Area Chair

---

### Official Review · Reviewer_yuhH · 2025-06-30

**Clarity:** 4
**Significance:** 3
**Originality:** 3
**Rating:** 5
**Confidence:** 4

**Summary:**

Paper Motivation:

The paper addresses the task of interacting with 3d scenes with natural language queries.
For this the scene must be augmented with language features (e.g. CLIP features). Previous work LangSplat allows to embed CLIP features to scenes reconstructed with 3D Gaussian Splatting.
But LangSplat does not achieve real-time performance, limiting its applications.

Contribution:

The authors propose to use codebook learning to improve rendering speed of 3DGS with language features:
- during reconstruction a global codebook is learned
- each Gaussian has learnable coefficients for the entries in the codebook
- only top k coefficients are used to increase performance
- during rendering the coefficients are rasterized into a feature image
- per pixel CLIP features are computed as linear combination of feature image and codebook

Results:

Authors show that their method archives real time rendering with SOTA accuracy on multiple datasets.
Results are compared to existing methods and an ablation study is performed to measure the effect of introduced hyper-parameters.

**Questions:**

The paper is well written and in a very good state. My main concern is the related work section which seams like an arbitrary list of work related to 3DGS in some parts and misses related work which also aims to speed up rendering of 3DGS.

If the authors address my concerns about the related work I will increase my score.

**Ethical Concerns:**

["NO or VERY MINOR ethics concerns only"]

**Final Justification:**

The authors have adequately addressed my concerns regarding the lack of related work and I therefore raise my score to accept (5).

**Limitations:**

Limitations are adequately addressed.

**Paper Formatting Concerns:**

-

**Quality:**

3

**Strengths And Weaknesses:**

Strengths:

- paper is well written and has an easy to follow clear structure
- graphics are well made and help to better understand the method
- addresses major limitation (performance) of existing methods without reducing accuracy (!!!)
- novel approach for accelerating rendering of high dimensional features
- authors promise to release code

Weaknesses:
- related work seems minimal / short and can be improved
- Paper focuses on making 3DGS rendering faster, related work should include papers with similar goals.
- specifically, other works  (e.g. Compressed 3D Gaussian Splatting) have used codebooks to accelerate 3DGS rendering. The authors discuss this in the appendix but this should definitely be moved to related work in the main paper.
- training is 3x slower compared to LangSplat. IMO this should be mentioned with numbers in the paper (not just reference to appendix)

---

> ### Author Rebuttal · Authors · 2025-07-31
>
> Thank you for the constructive feedback and the positive assessment of our work! Below, we detail our responses to the review questions.
>
> **Q1: My main concern is the related work section which seams like an arbitrary list of work related to 3DGS in some parts and misses related work which also aims to speed up rendering of 3DGS.**
>
> Thank you for the helpful suggestion. We will revise and expand the Related Work section in the main paper accordingly. Specifically, we will reorganize the current content and include a dedicated paragraph discussing 3D Gaussian Splatting acceleration, including relevant works such as Compressed 3D Gaussian Splatting that also adopt codebook-based strategies for faster rendering. We will include the following discussion in the related work section.
>
>
> >To reduce computational overhead and improve rendering speed, recent works have proposed various strategies to accelerate 3D Gaussian Splatting (3DGS). One line of work compresses the Gaussian representation using vector quantization and codebooks. For instance, Compressed 3D Gaussian Splatting [1] adopts sensitivity-aware clustering and quantization-aware training to compress both directional colors and Gaussian parameters, achieving up to 31× compression and 4× rendering speedup. CompGS [2] similarly applies K-means-based quantization to achieve 40–50× storage reduction and 2–3× faster rendering with minimal quality loss.
>
> >Other approaches focus on structural regularization or architectural simplification. Self-Organizing Gaussian Grids [3] reorganize Gaussian parameters into spatially coherent 2D grids, reducing redundancy via local homogeneity. LightGaussian [4] uses pruning based on global significance and vector quantization, achieving 15× compression and 200+ FPS rendering. [5] proposes to reduce the number of Gaussians via a learnable mask and introduces compact view-dependent color encoding, achieving a 25× reduction in storage. Techniques like Speedy-Splat [6] and SORT-Free Splatting [7] improve runtime performance through optimized rasterization and differentiable approximations of alpha blending.
>
> >In contrast to these works that focus on accelerating RGB-based scene rendering, our method tackles the unique challenge of handling high-dimensional Gaussian features (e.g., 512D). Extending prior compression techniques [1, 4] to our setting would require learning and quantizing such high-dimensional attributes per Gaussian, leading to excessive memory usage. Moreover, directly rendering with full 512D features remains computationally demanding. In contrast, our proposed method circumvents the need to directly learn and render high-dimensional features for 3D Gaussian points.
>
> [1] Niedermayr, Simon, Josef Stumpfegger, and Rüdiger Westermann. "Compressed 3d gaussian splatting for accelerated novel view synthesis." CVPR 2024.
>
> [2] Navaneet, K. L., et al. "Compgs: Smaller and faster gaussian splatting with vector quantization." ECCV 2024.
>
> [3] Morgenstern, Wieland, et al. "Compact 3d scene representation via self-organizing gaussian grids." ECCV 2024.
>
> [4] Fan, Zhiwen, et al. "Lightgaussian: Unbounded 3d gaussian compression with 15x reduction and 200+ fps." NeurIPS 2024.
>
> [5] Lee, Joo Chan, et al. "Compact 3d gaussian representation for radiance field." Proceedings of the CVPR 2024.
>
> [6] Hanson, Alex, et al. "Speedy-splat: Fast 3d gaussian splatting with sparse pixels and sparse primitives." CVPR 2025.
>
> [7] Hou, Qiqi, et al. "Sort-free gaussian splatting via weighted sum rendering." ICLR 2025.
>
>
>
> **Q2: Training numbers should be mentioned in the paper (not just reference to appendix)**
>
> Thanks for the suggestion. We will mention it and provide a detailed analysis in the main paper in the revised version.

---

### Official Review · Reviewer_Vun8 · 2025-07-02

**Clarity:** 4
**Significance:** 3
**Originality:** 3
**Rating:** 5
**Confidence:** 4

**Summary:**

Summary:
- In this work authors build LangSplatv2, which results in significant (more than 450 times!) speedup of the prior work LangSplat.
- The primary contribution is improving the speed bottleneck of the prior work, which is decoding the low dimension textual embedding to high dimension (in CLIP space).
- The speed up comes from using a sparse (K-dim) 3D coefficient field and a global codebook (of total L basis vectors) to represent the language field.
- To convert the sparse coefficients to dense embedding, authors use top-K pruning and softmax normalization.
- Authors conduct exhaustive experiments comparing performance against baselines and several scenes, as well as benchmark their timings.

**Questions:**

- Will you open-source the code? Since majority gains in this work come from low-level optimizations (parallelizing alpha-blending, etc) I believe this would be very helpful.

**Ethical Concerns:**

["NO or VERY MINOR ethics concerns only"]

**Final Justification:**

I thank the authors for providing detailed explanation, and clarifications.

I will keep my original score of Accept for this work.

**Limitations:**

Yes.

**Paper Formatting Concerns:**

Typo in line 298 (‘totoal’)

**Quality:**

3

**Strengths And Weaknesses:**

Strengths:
- I believe this work takes the next right step in optimizing for the speed-up of a pipeline like LangSplat, which can enable several open-ended downstream applications.
- The gains are impressive - both in terms of speed-up, 3D localization and semantic segmentation performance as well.
- Experiments are thorough and compare against many existing baselines, ablation are comprehensive including timing studies.
- Paper is written well, and the figures are neat and helpful.

Weaknesses:
- The only major issue is - I was hoping to find videos in the supplementary, but couldn’t find any - it would’ve been helpful to see the visuals and qualitatively compare against said baselines.
- Could you demonstrate visually where the LangSplat vs LangSplatv2 performance gap in the Kitchen scene appears from? I want to understand the claim on L298 better, from Figure 5. LangSplatV2 appears to be better.
- The training time seems to have tripled compared to LangSplat, I wonder if the authors tried any workarounds for this?

---

> ### Author Rebuttal · Authors · 2025-07-31
>
> Thank you for the constructive feedback and the positive assessment of our work! Below, we detail our responses to the review questions.
>
>
> **Q1: Video Demo**
>
> Thank you for the suggestion. Due to NeurIPS rebuttal guidelines, we are unable to upload images or videos during the rebuttal phase. However, we will publicly release video demos that show visual comparisons with the baselines upon acceptance.
>
> **Q2: Performance Gap in the Kitchen Scene**
>
> Thanks for the question. Actually LangSplatV2 attains a more accurate 3D language field in the Kitchen scene.
>
> To better understand the performance gap, we encourage referring to Table 2 of our main paper, where LangSplatV2 significantly outperforms LangSplat in the 3D Semantic Segmentation task (59.1% vs. 44.5%). This task provides a more reliable and comprehensive assessment of the learned 3D language field, as it considers the overall semantic consistency of the entire 3D space rather than relying on a single point prediction.
>
> It's important to note that the 3D Object Localization task uses the location with the maximum similarity as the predicted object center, which can be disproportionately affected by noise and local artifacts. In the Kitchen scene specifically, this sensitivity—combined with the limited number of test samples—results in high variance. In fact, when we ran multiple evaluations on the Kitchen scene, the average localization accuracy of LangSplat was 79.5%, compared to 86.4% for LangSplatV2, demonstrating the superiority of LangSplatV2.
>
> Finally, Figure 5 qualitatively validates that LangSplatV2 exhibits more semantically accurate and spatially coherent activations, reinforcing that it indeed learns a better 3D language field in the Kitchen scene.
>
>
> **Q3: Workarounds for the Training Time**
>
> Thank you for your question. The increased training time compared to LangSplat primarily stems from the need to learn high-dimensional 3D semantic fields in LangSplatV2, which naturally incurs more computational overhead. We acknowledge this trade-off and have actively explored ways to mitigate it.
>
> Among the techniques we tested, one of the most effective was the initialization strategy for the global codebook. Instead of using random initialization, we first extract all semantic features from the input 2D images and perform clustering. The resulting cluster centers are then used to initialize the codebook. This initialization significantly improves convergence speed.
>
> As shown in the table below, training with this initialization (denoted as “w init”) achieves much lower training loss at early iterations compared to the version without initialization (“w/o init”). Specifically, with only 10K iterations, the “w init” version reaches a similar loss level as the “w/o init” version after 30K iterations. As a result, we were able to reduce LangSplatV2’s training iterations from 30K (used in LangSplat) to just 10K, effectively narrowing the training time gap. We recognize that training efficiency remains an important concern and plan to further investigate strategies to accelerate training. We believe this will be a valuable and interesting direction for future work.
>
> |    Training Loss     | 1k   | 2K   | 4K   | 6k   | 8k   | 10k  | 15k  | 20k  | 25k  | 30k  |
> |---------|------|------|------|------|------|------|------|------|------|------|
> | w init  | 0.191| 0.181| 0.177| 0.172| 0.172| 0.168|  -   |  -   |  -   |  -   |
> | w/o init| 0.260| 0.210| 0.195| 0.189| 0.181| 0.175| 0.170| 0.169| 0.169| 0.168|
>
> **Q4:Will you open-source the code?**
>
> We commit to releasing the complete source code, training scripts, and model weights to ensure full reproducibility upon acceptance.
>
> **Q5:Typo**
>
> Thanks! We will fix it in the revised version.

---

> > ### Comment · Reviewer_Vun8 · 2025-08-05
> >
> > Thanks for providing detailed explanation, and clarifications. I will keep my score of Accept for this work.

---

> > > ### Author Response · Authors · 2025-08-05
> > > **Thank you**
> > >
> > > Thank you for your reply and your positive feedback. We appreciate it very much!

---

### Official Review · Reviewer_T9NY · 2025-07-03

**Clarity:** 3
**Significance:** 3
**Originality:** 3
**Rating:** 4
**Confidence:** 3

**Summary:**

The paper presented LangSplatV2, an approach for high-dimensional language feature splatting in 3D open-vocabulary querying, aiming to address the speed limitations of LangSplat. By identifying the decoding stage in LangSplat to be the bottleneck, a 3D sparse coefficient field field is proposed to represent the language embedding of each Gaussian point, coupled with an efficient scheme to learn such coefficients, LangSplatV2 obtained significant speed up compared to LangSplat.

**Questions:**

What are input image resolutions of datasets that evaluation was conducted? How sensitive is LangSplatV2 regarding this?

**Ethical Concerns:**

["NO or VERY MINOR ethics concerns only"]

**Final Justification:**

Given the authors' rebuttal has addressed my questions, I maintain my rating "Borderline Accept".

**Limitations:**

Yes

**Paper Formatting Concerns:**

No concern regarding paper formatting.

**Quality:**

3

**Strengths And Weaknesses:**

[Strengths]

1) Thorough analysis performed to identify computation bottleneck in LangSplat.
2) Representing each Gaussian point with a sparse coefficient directly learned from codebook in 3D space is novel and effective.
3) Extensive experiments demonstrated that LangSplatV2 achieved solid improvements across a number of tasks compared to previous methods.

[Weakness]

I do not see major weakness of the proposed LangSplatV2. Although if runtime analysis when compared with other methods can be included, it would give a more wholistic picture. It would also be good to see code open-sourced.

---

> ### Author Rebuttal · Authors · 2025-07-31
>
> Thank you for the constructive feedback and the positive assessment of our work! Below, we detail our responses to the review questions.
>
> **Q1: Runtime Analysis**
>
> Thanks for the suggestion. We provide a detailed runtime comparison in the table below.
>
>
> | Method | Render(ms) | Decode(ms) | Post(ms) | Total(ms) | FPS |
> |----------|----------|----------|-------|-----|-----|
> | GS-Grouping  | 4.4      | 0.5       | 659.5 | 664.4 | 1.5 |
> | LEGaussian   | 9.8      | 4.3       | 22.6 | 36.7 | 27.2 |
> | LangSplat*    | 2.0      | 83.1      | 0.5 | 85.6 | 11.7 |
> | **LangSplatV2**|**2.0**|**0.1**|**0.5**|**2.6**|**384.6**|
>
> The results are obtained on the LERF dataset with one A100 GPU. It can be observed that LangSplatV2 achieves the fastest speed at every stage, leading to a substantial improvement in inference efficiency. Furthermore, Tables 2, 3, and 4 in the main paper demonstrate that LangSplatV2 attains state-of-the-art performance across multiple datasets. In summary, LangSplatV2 not only achieves superior or competitive query accuracy but also exhibits significantly higher speed.
>
>
> **Q2:It would also be good to see code open-sourced.**
>
> Yes, we will release the complete source code, training scripts, and model weights to ensure full reproducibility upon acceptance.
>
>
>
>
> **Q3:Input Image Resolutions**
>
>
>
> Thank you for the question. The input image resolutions used in our evaluation are: 730×980 for LERF, 1080×1440 for 3D-OVS, and 1080×1620 for Mip-NeRF360. These datasets cover a wide range of resolution settings. As shown in the table below, our method LangSplatV2 consistently outperforms prior approaches such as GS-Grouping and LEGaussian in terms of segmentation accuracy, query speed, and memory efficiency across all these datasets. Despite the varying input resolutions, LangSplatV2 maintains real-time query performance (as low as 2.6 ms on LERF, 4.8 ms on 3D-OVS, and 3.8 ms on Mip-NeRF360) while also achieving the highest Segmentation IoU on all three benchmarks. This demonstrates that our method is robust to input resolution changes and can deliver high-quality, efficient performance across different scene configurations.
>
>
> | LERF(730 $\times$ 980) | Seg(IoU) |Query Time(ms)|Memory(GB)|
> |-------------|---------------------|------|-----|
> | GS-Grouping | 36.6 | 664.4 | 7.3  |
> | LEGaussian  | 24.6 | 36.7 | 8.2 |
> | **LangSplatV2** | **59.9** | **2.6** | **7.2** |
>
> | 3D-OVS(1080 $\times$ 1440) | Seg(IoU) |Query Time(ms)|Memory(GB)|
> |-------------|---------------------|------|-----|
> | GS-Grouping | 87.7 | 1016.4 | 7.6  |
> | LEGaussian  | 88.5 | 59.6 | 10.5 |
> | **LangSplatV2** | **94.6** | **4.8** | **11.6** |
>
>
> | Mip-NeRF360(1080 $\times$ 1620) | Seg(IoU) |Query Time(ms)|Memory(GB)|
> |-------------|---------------------|------|------|
> | GS-Grouping | 49.2 | 1120.2 | 12.0  |
> | LEGaussian  | 29.1 | 58.0 | 14.0 |
> | **LangSplatV2** | **69.4** | **3.8** | **16.9** |

---

> > ### Author Response · Authors · 2025-08-05
> > **Looking forward to your feedback**
> >
> > Thanks again for your valuable advice and supportive comments! We have responded to your initial comments. We are looking forward to your feedback and will be happy to answer any further questions you may have.

---

> > ### Comment · Reviewer_T9NY · 2025-08-05
> >
> > Thanks for addressing my questions. I will keep my initial rating.

---

### Comment · Area_Chair_M2Ya · 2025-08-04

Dear Reviewers,

Thank you all for your thorough reviews. The authors have provided detailed responses to your concerns.

The discussion period ends August 6. Please engage with the authors and each other to clarify any remaining concerns.

Best regards, Area Chair

---

### Note · Authors · 2025-08-12

We sincerely thank all reviewers for their time, efforts, and constructive feedback throughout the review process. In the initial review stage, our submission received consistent positive evaluations from all four reviewers (three ***Borderline Accept*** and one ***Accept***).

During the rebuttal and discussion stages, we clarified several important points and conducted additional analyses/experiments:

1.	**Comprehensive runtime analysis:** We performed stage-wise comparisons with competing methods and evaluated performance across varying image resolutions, showing that our method consistently outperforms all competitors while maintaining truly real-time inference speed.
2.	**Training cost clarification:** We clarified the training cost in more detail and showed that initializing the codebook via 2D language feature clustering can accelerate training.
3.	**Reproducibility verification:** We repeated experiments multiple times, demonstrating both high average performance in all scenarios and small variance in most cases, highlighting the method’s stability and effectiveness.
4.	**Extended related work:** We broadened the discussion on 3D Gaussian compression and acceleration, emphasizing our unique contribution to reconstructing and rendering high-dimensional semantic features.

During the discussion phase, Reviewer T9NY and Reviewer Vun8 explicitly confirmed that their concerns had been addressed and maintained their positive scores. Besides, the remaining reviewers did not raise any further questions or concerns.

We once again thank the reviewers and the Area Chair for their valuable feedback and thoughtful engagement.

---

### Decision · Program_Chairs · 2025-09-17

**Decision:**

Accept (poster)

**Comment:**

This paper achieves 47x speedup over LangSplat (476 FPS) by replacing the MLP decoder with a sparse coefficient field and global codebook. Four reviewers with three engaged post-rebuttal. One reviewer raised the score to 5, while another remained unresponsive. The proposed method demonstrates significant practical impact, impressive speedup that maintains accuracy, and thorough experiments on LERF, 3D-OVS, and Mip-NeRF360 datasets. Authors addressed concerns, including theoretical novelty (clarified key differences from LEGaussian), 3x training cost (optimization reduces iterations from 30k to 10k), and statistical validation (additional experiments show 94.6 mIoU) during the rebuttal. Despite limited theoretical novelty, the substantial engineering contribution enables real-time applications. Therefore, the AC recommends Acceptance as a poster paper.